# Quantifying the impact of COVID-19 on e-bike safety in China via multi-output and clustering-based regression models

Xingpei Yan[1,2], Zheng Zhu[3]*

**1** School of Automobile, Chang'an University, Xi'an, P.R. China, **2** Department of Traffic Policy Planning Research, Research Institute for Road Safety of Ministry of Public Security, Beijing, P.R. China, **3** Department of Civil and Environmental Engineering, the Hong Kong University of Science and Technology, Hong Kong, P.R. China

* zhuzheng@ust.hk

**Data Availability Statement:** Three data sources are used in this paper: 1) The province-level and monthly socioeconomic data of China, which is public (open access) and can be downloaded from the link http://www.stats.gov.cn/. The data includes socioeconomic information such as seasonal total

## Abstract

The impacts of COVID-19 on travel demand, traffic congestion, and traffic safety are attracting heated attention. However, the influence of the pandemic on electric bike (e-bike) safety has not been investigated. This paper fills the research gap by analyzing how COVID-19 affects China's e-bike safety based on a province-level dataset containing e-bike safety metrics, socioeconomic information, and COVID-19 cases from 2017 to 2020. Multi-output regression models are adopted to investigate the overall impact of COVID-19 on e-bike safety in China. Clustering-based regression models are used to examine the heterogeneous effects of COVID-19 and the other explanatory variables in different provinces/municipalities. This paper confirms the high relevance between COVID-19 and the e-bike safety condition in China. The number of COVID-19 cases has a significant negative effect on the number of e-bike fatalities/injuries at the country level. Moreover, two clusters of provinces/municipalities are identified: one (cluster 1) with lower and the other (cluster 2 that includes Hubei province) higher number of e-bike fatalities/injuries. In the clustering-based regressions, the absolute coefficients of the COVID-19 feature for cluster 2 are much larger than those for cluster 1, indicating that the pandemic could significantly reduce e-bike safety issues in provinces with more e-bike fatalities/injuries.

## Introduction

Since the declaration of the COVID-19 pandemic by the World Health Organization (WHO) in February 2020 [1], the implementation of lockdown policies has been a global trend. The accompanying non-pharmaceutical interventions, such as travel restrictions and social distance policies, raise negative influences on productivity, cause large social costs, and shock the global financial market and social economy. Furthermore, the COVID-19 pandemic has been reshaping the way people travel such that human mobility patterns in the coming years could be notably different from the past. As a result, people are paying more and more attention to the impact of COVID-19 on travel demand, traffic congestion, and traffic safety. Evidence has

GDP, electric power consumption, and the profit from express and logistics. 2) The province-level COVID-19 data of China, which is public (open access) and can be downloaded from the link https://en.wikipedia.org/wiki/Statistics_of_the_COVID-19_pandemic_in_mainland_China. The data includes the number of COVID-19 cases, recoveries, and deaths for each month and province. 3) The monthly and province-level China e-bike safety data is not public. The dataset is owned by the Research Institute for Road Safety (RIRS) of the China Ministry of Public Security (MPS). The data includes e-bike safety information, such as the number of fatalities, the number of injuries, property damage, the distribution of accident causes. The authors of this paper do not have special privileges in accessing the data. For other researchers interested in this dataset, please contact zhuxinyu@122.cn (Mr. Xinyu Zhu, assistant researcher at RIRS, "122.cn" is the official email domain for the China government) for permission/availability of the dataset; any academic use of the dataset will be welcomed, but it could take weeks for processing/delivery of the data due to authority processes from China MPS. Since data 3) is owned by the RIRS, we cannot release the original raw data. However, we have contacted the data owner who allowed us to publicize some processed data used in this paper. In this manner, we share the data from 2019 to 2020 as a minimal anonymized data set. Please find the data and introduction in the Supporting Information.

**Funding:** The research is supported by Central Public-Interest Scientific Institution Basal Research Fund (Grant No. 11104100000180001210102).

**Competing interests:** No authors have competing interests.

been found all around the world on how the pandemic affects urban mobility. For instance, the vehicle miles traveled and driving frequency dropped over 30% in Alabama, US [2], the speed of traffic flow increased due to lower congestion level in New York City [3], and the number of car crashes dropped [4], the usage of public transit fell at the beginning of the pandemic and then recovered in the UK [5], and people made fewer walk/bike trips in densely populated cities but more in less densely populated cities in the US [6]. Yet, no effects have been conducted on the influence of COVID-19 on an indispensable mobility mode—the electrical bike (e-bike).

The e-bike, a type of powered two-wheeled vehicle that is inexpensive, fast, and environmentally friendly (compared with motor vehicles), is widely used in many places such as Australia [7], Europe [8], the US [9], and China [10]. With a tremendous grown in usage over the past two decades, the e-bike has become a major nonmotorized travel mode in China. In 1998, there were only several thousand e-bikes produced and sold in China, and this number grew to over 20 million in 2009. From 2013, China became the largest e-bike producer and exporter. Now the yearly production and sales are over 350 million. Generally, e-bikes are popular in China because they allow people to commute longer distances and are serving as a high-quality mobility alternative to public transit and conventional bikes. Furthermore, China has been promoting the shared economy, and most on-demand (food) delivery men who are working for companies such as Meituan and Ele.me rely on e-bikes to finish jobs on time. However, e-bikes also raise road safety issues because of the fast speed and the less protected riding environment compared with cars. Researchers have found that many e-bike riders have risky behaviors such as speeding, riding on motorized roadways, not wearing helms, or disobeying traffic signals [11–14]. As a result, the number of fatalities/injures in e-bike accidents has become a major component in road traffic fatalities/injures in China; in 2019, nearly 30% of road traffic fatalities/injures were caused by e-bike accidents (see **Fig 1** that illustrates the number of fatalities/injures of different travel modes in 36 major cities in mainland China). The pros and cons of e-bikes could be more significant during the COVID-19 pandemic. On one hand, e-bikes naturally meet the requirement of social distance for preventing the epidemic from spreading and would be more preferable than public transit and cars among commuters. On the other hand, high e-bike usage during the pandemic could continuously cause traffic safety issues.

This paper fills a research gap by quantifying the impact of the COVID-19 pandemic on e-bike safety in China. We conduct two analyses based on a province-level dataset with e-bike safety, COVID-19, and socioeconomic metrics from 2017 to 2020. First, we examine the overall impact of COVID-19 on e-bike safety at the country level. Second, we conduct clustering-based regressions to investigate the heterogeneous influence of COVID-19 and other explanatory variables on e-bike safety conditions at the province level. The results demonstrate the high relevance between COVID-19 and e-bike fatalities/injuries in China. On the province level, the pandemic has a larger negative impact on the number of e-bike fatalities/injuries for provinces with more serious e-bike safety issues. Since e-bikes provide a comfortable mode for preventing the epidemic from spreading, our analyses provide good references for the study of mobility and traffic safety during-and-post the pandemic.

## Literature review

Before discussing the data, methodology, and analysis results, we present a review of the impact of COVID-19 on travel demand and traffic safety, as well as the key features/covariates used for traffic safety regression studies. The review identifies the contributions of this paper as a supplement to the existing literature on COVID-19 and traffic safety-related studies.

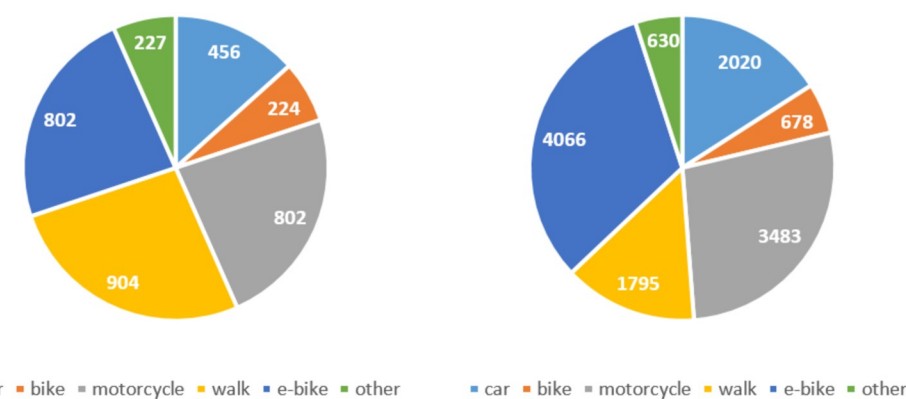

**Fig 1. Statistics of traffic accidents in 36 major cities in mainland China in 2019.** (a) The number of fatalities. (b) The number of injuries.

Since the breakout of COVID-19 in Hubei, China in late 2019, there have been three worldwide infection waves of the epidemic: 1) the first wave peaked between early spring and summer in 2020 when countries started to implement lockdown strategies; 2) the second wave witnessed the rebounding of infection during October to the end of 2020 as some countries relaxed social distance policies; 3) the third wave began in early 2021 along with the start of vaccination campaigns since some countries loosed lockdowns and travel restrictions [15]. The threat from the virus and lockdown policies have significantly changed people's travel demand and travel patterns. The most heavily damaged transportation sector is air transport, which suffered from a decline of 50% seats (around 2.9 billion passengers and 390 billion USD) [16]. Similarly, rail transport, another essential long-distance travel mode, has experienced 20% to 30% annually passenger loss in different regions such as Europe, the US, and Asia [17, 18]. Apart from long-distance mobility, it is worth noting that both land-borne and maritime freight transport have met certain degrees of financial crisis in the phase of the pandemic; while e-commerce companies, which focus on business-to-consumer sections, such as UPS, FedEx, and Amazon, became the winner [15]. Overall, the findings indicate a notable decline in people's travel/social/shopping activities.

We examine how the pandemic has been changing people's daily travel patterns in terms of the overall trip demand and the usage of different ground transport modes. Based on long-term mobile device location data in the US, the University of Maryland research team showed that the pandemic itself has dramatically reduced people's number of trips and trip mileages, especially in states in a serious infection situation; while the pure impact of lockdown policies signed by governments was much smaller [19–21]. Using the same dataset, Xiao et al. found that people's trip duration has notably decreased after the national call [22]. Similar evidence on the decline of travel frequency and duration was found in other regions, such as Asia and Europe [23, 24]. Besides changes in travel frequency and duration, it was observed that people tend to choose private cars or bikes rather than the subway, buses, taxis, or ride-hailing services to stay away from the gathering in many places, such as Germany, Canada, Scotland, and China [25–28].

In addition to reshaping people's travel demand and habits, the pandemic and lockdown policies also affect traffic safety. It was found that the number of roadway accidents decreased significantly in the US [29–31]; however, the fatality rate during traffic accidents increased by around 14% in early 2020 compared with 2019 [32]. Using traffic flow and incident data in Greece, Katrakazas et al. found that the lockdown policy caused more vehicle accidents since

people drove faster with fewer vehicles on roadways [33]; speeding behaviors during the epidemic were also found in the UK and France [34]. The ETSC reported a notable decrease in the number of incidents/fatalities during the pandemic in some European countries, such as Spain, Italy, Finland, and Germany, which was largely due to the reduction of trips [33, 34].

According to the aforementioned review, there are still insufficient studies about the impact of COVID-19 on roadway safety; and most safety analyses were conducted based on car-related crashes, while nonmotorized travel modes (e.g., walk, bike, and e-bike) have drawn little attention. It was estimated that cyclists and pedestrians would enjoy a much safer trip since the pandemic has led to certain reductions in vehicle volume on roadways [35]. With e-bike accident data before and during the pandemic, we try to fill a research gap by analyzing how COVID-19 impacts e-bike safety in China. To conduct a solid empirical study, we fuse the accident dataset with public socioeconomic features that are highly correlated with traffic/bike/e-bike safety based on the literature. For example, population and population density generally indicate the overall travel demand, and they can be positively related to the number of accidents [36, 37], GDP and economic-related features reflect people's activity and travel frequency [36, 37], income tends to have positive effects on car travel demand [38], age group and gender are associated with travel mode and driving/riding behavior [39, 40], urbanization level partially shows travel density in the study area [37, 41].

## E-bike safety data of mainland China

The dataset used in this case study is fused based on province-level e-bike accident statistics from the Research Institute for Road Safety of Ministry of Public Security, province-level socioeconomic metrics from the China National Bureau of Statistics, and province-level COVID-19 cases data from Wikipedia (please refer to the S1 File online for the web links of the dataset). Detailed e-bike accident and socioeconomic information for all the 31 provinces and municipalities from the year 2017 to 2020 is recorded in the dataset, including the monthly number of fatalities, monthly number of injuries, monthly property damage, the distribution of accident causes, annual population, seasonal GDP, monthly electric power consumption, monthly profit from express and logistics, annual urbanization rate, average annual income, and annual age distribution. There are 1,488 data records in total for 48 months and 31 provinces/municipalities. The map of provinces is shown in **Fig 2** (created based on the USGS National Map Viewer): 1 Beijing, 2 Tianjin, 3 Hebei, 4 Shanxi, 5 Nei Menggu, 6 Liaoning, 7 Jilin, 8 Heilongjiang, 9 Shanghai, 10 Jiangsu, 11 Zhejiang, 12 Anhui, 13 Fujian, 14 Jiangxi, 15 Shangdong, 16 Henan, 17 Hubei, 18 Hunan, 19 Guangdong, 20 Guangxi, 21 Hainan, 22 Chongqing, 23 Sichuan, 24 Guizhou, 25 Yunnan, 26 Tibet, 27 Shaanxi, 28 Gansu, 29 Qinghai, 30 Ningxia, 31 Xinjiang.

**Table 1** presents the statistics of variables used in this paper. Note that we are not fitting a panel data model so that the annual/monthly linear trend is not considered. Property damage is not considered in our study because the costs/prices of e-bikes are much lower than motor vehicles. The majority of statistics are summarized based on province and month except for variables "seasonal GDP", "population", "urbanization rate", "average annual income", "percentage of age group 1" and "percentage of age group 2". For 3 months in the same season, the values of GDP are identical to the seasonal GDP of the province. For instance, if the GDP for Beijing during spring 2017 is 500 billion CNY, then we set the GDP for Beijing in January, February, and March to be 500 billion CNY. It is unnecessary to divide the value by 3 (months) since we are developing linear regression models. For 12 months in the same year, the values of average income, urbanization rate, population, and age distributions are the same for specific provinces and are regarded as annual fixed effects for regressions. The total numbers of e-

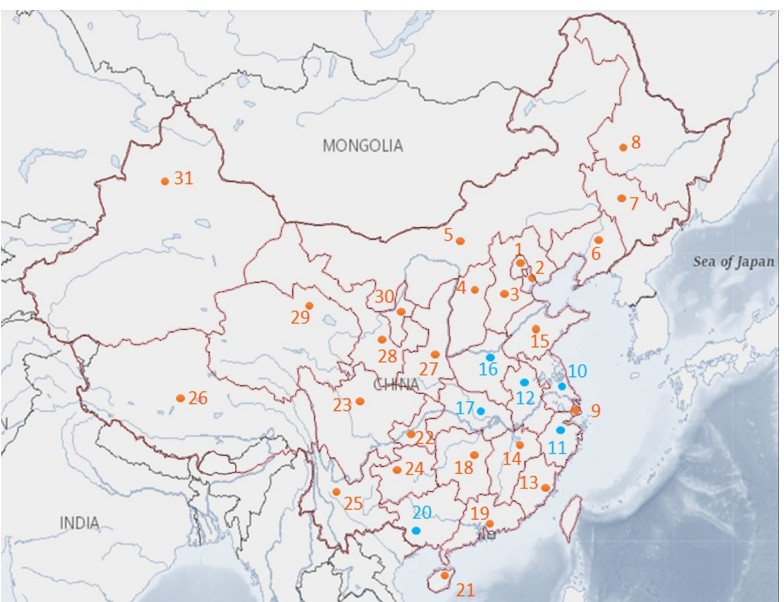

**Fig 2. Map of provinces and municipalities in mainland China.**

bike fatalities and injuries during the four years in China are 11,137 and 88,331, respectively. Generally, injuries are recorded by the police upon incident calls, but provinces may have different counting standards. For 840 out of the 1,488 data records (56.5%), the number of e-bike fatalities is less or equal to 5 per province per month; for 54 out of the records (3.6%), the number of fatalities is over 30 per province per month and such records are observed in Jiangsu, Zhejiang, and Guangxi (point 10, 11, and 20 in **Fig 2**); the highest number of fatalities is 62, which happened in Jiangsu. The spatial heterogeneity of e-bike fatalities could be caused by the differences in policy supports for using e-bikes, economic levels, and people's travel preferences. The maximum number of COVID-19 cases is 59,754 (reported by Hubei province in February 2020), which is extremely larger than any other record (the second large record is 7,153 for Hubei in January 2020). To develop sound linear models that ascertain the true effect of COVID-19, we make a nonlinear transformation of the number of COVID-19 cases by

**Table 1. Variables and statistics.**

| Variables | Statistics | | | |
|---|---|---|---|---|
| | **Min.** | **Med.** | **Max.** | **Avg.** |
| # of fatalities for e-bike accidents | 0.000 | 5.000 | 62.00 | 7.481 |
| # of injures for e-bike accidents | 0.000 | 30.00 | 520.0 | 59.37 |
| # of COVID-19 cases | 0.000 | 0.000 | 59754.0 | 58.52 |
| log-transformed # of COVID-19 cases | 0.000 | 0.000 | 4.776 | 0.232 |
| population (million people) | 3.490 | 39.38 | 126.0 | 45.31 |
| seasonal GDP (trillion CNY) | 0.027 | 0.595 | 3.236 | 0.760 |
| profit from express/logistics (billion CNY) | 0.009 | 0.663 | 22.25 | 1.834 |
| power consumption (billion KWH) | 0.406 | 14.85 | 116.9 | 18.74 |
| urbanization rate | 29.80 | 60.03 | 86.62 | 60.46 |
| average annual income | 15.46 | 25.06 | 72.23 | 29.22 |
| percentage of age group 1 (15 to 64) | 63.37 | 71.16 | 78.81 | 71.24 |
| percentage of age group 2 (over 64) | 5.670 | 11.97 | 17.42 | 11.78 |

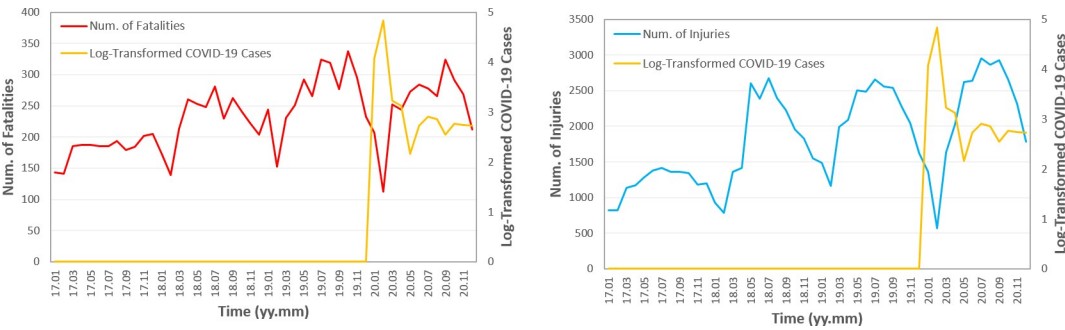

**Fig 3. The trend of e-bike safety in China.** (a) the number of fatalities. (b) the number of injuries.

using $\log_{10}(x_t+1)$, where $x_t$ denotes the monthly province-level confirmed cases; therefore, the lowest value of "log-transformed # of COVID-19 cases" equals zero for 0 cases, and the maximal value equals 4.776 with 59,754 cases. For all these variables, the average values are greater than the medium values, indicating that there are "metropolitan" provinces/municipalities in China. The large range of socioeconomic variables, such as 0.027 T to 3.236 T CNY for seasonal GDP and 0.406 to 116.921 B KWH for power consumption, 15.46 K CNY to 72.23 K CNY for average annual income, and 29.80% to 86.62% for urbanization rate, illustrates the unbalance in development, which largely depends on the location of the provinces/municipalities and the national economic policy.

Fig 3 displays the trend of e-bike safety in China from 2017 to 2020; the time series data is obtained by summing monthly numbers of fatalities and injuries across the 31 provinces/municipalities. We also show the log-transformed total number of COVID-19 cases (i.e., $\log_{10}(\sum_{j=1}^{31} x_t^j + 1)$, where $x_t^j$ denotes the number of COVID-19 cases for province $j$ during month $t$), which refers to the right vertical axis of the figures. As depicted in the figures, the numbers of fatalities and injuries show a notable yearly (i.e., 12-monthly) periodic trend; for each year, the number of fatalities/injuries researched the bottom during January and February due to cold weather and the Chinese New Year. Furthermore, the number of fatalities/injuries gradually increased from 2017 to 2019; while the outbreak of COVID-19 in January and February 2020 remarkably slowed down the increase of e-bike fatalities/injuries cases.

## Methodology

To provide an in-depth examination of the impact of COVID-19 on e-bike safety in China, we perform an aggregate country-level analysis and a province-level analysis. In the former task, we develop two multi-output regression models based on country-level time-series data: one considers but the other ignores the COVID-19 variable; the similarity/difference of results and the goodness-of-fit tell whether COVID-19 has a major/minor influence on e-bike safety. The latter task utilizes clustering-based multi-output regression models, in which provinces with a similar data pattern are grouped into clusters, and regressions are conducted within each cluster; therefore, we ascertain the heterogeneity of COVID-19's impacts across different clusters. All statistical analyses were carried out using the programming language python.

### Multi-output regression models

In the country-level analysis, we consider the following time series linear regression model:

$$\boldsymbol{y}_t = \boldsymbol{x}_t \boldsymbol{w} + \boldsymbol{e}_t \tag{1}$$

where $t \in \{1, 2, \ldots, T\}$ denotes the time index, $T$ is the number of total time periods, $\boldsymbol{x}_t = (x_{t,1}, x_{t,2}, \ldots x_{t,M})$ is the vector of $M$ explanatory variables and $\boldsymbol{y}_t = (y_{t,1}, y_{t,2}, \ldots y_{t,D})$ denotes the vector of $D$ dependent variables, $\boldsymbol{w}$ denotes a $M \times D$ coefficient matrix, and $\boldsymbol{e}_t$ denotes a $1 \times D$ residual vector.

We call the model in Eq (1) a multi-output regression model since its dependent variable has multiple dimensions, which has been used in transportation regression problems such as travel behavior prediction [42] and traffic safety analyses [36]. Comparing with fitting several single-output regression models, the training of a multi-output model takes advantage of correlations among the dependent variables and improves prediction accuracy, especially when dependent variables are significantly correlated with each other [36, 43]. In this paper, the aforementioned correlations are modeled via the regularized linear structure that utilizes the following objective function:

$$\min_{\boldsymbol{w}} \frac{1}{2T} \sum_{t=1}^{T} \| \boldsymbol{y}_t - \boldsymbol{x}_t \boldsymbol{w} \|_F^2 + \lambda_1 \Phi(\boldsymbol{w}) + \lambda_2 \| \boldsymbol{w} \|_F^2 \qquad (2)$$

where term $\frac{1}{2T} \sum_{t=1}^{T} (\boldsymbol{y}_t - \boldsymbol{x}_t \boldsymbol{w})^2$ represents the least square loss for linear models, terms $\lambda_1 \Phi(\boldsymbol{w})$ and $\lambda_2 \| \boldsymbol{w} \|_F^2$ are used for model regularization to enhance the prediction accuracy and interpretability as well as reduce the collinearity between explanatory variables. $\lambda_1 \geq 0$ and $\lambda_2 \geq 0$ are complexity parameters for the cross-task regularization penalty $\Phi(\boldsymbol{w})$ and the $l_2$-norm penalty $\| \boldsymbol{w} \|_F^2$, respectively. The cross-task regularization penalty can take different forms, such as a $l_{2-1}$-norm ($\Phi(\boldsymbol{w}) = \| \boldsymbol{w} \|_{2,1}$), or a trace form ($\Phi(\boldsymbol{w}) = \mathrm{tr}(\boldsymbol{w})$, where $\mathrm{tr}(\cdot)$ denotes the trace operator of a matrix). Note that $\lambda_1 = 0$ indicates a ridge regression [44], $\lambda_2 = 0$ represents a lasso regression [45] and $\lambda_1 = \lambda_2 = 0$ means a conventional linear regression.

Based on the dataset discussed in **Section 3**, we train two multi-output regression models for comparison:

- **Non-COVID-19 Model**. In this model, $\boldsymbol{y}_t$ is a two-dimensional variable that contains the number of monthly fatalities and number of monthly injuries, $\boldsymbol{x}_t$ (7 dimensions) contains the 12-month lagged number of fatalities, the 12-month lagged number of injuries, seasonal GDP, express/logistics profit, power consumption, average annual income, and the percentage of age group 2. All these variables are summations of province-level metrics.

- **COVID-19 Model**. This model uses the same $\boldsymbol{y}_t$ as the non-COVID-19 model. Vector $\boldsymbol{x}_t$ (8 dimensions) includes all the variables in non-COVID-19, and we add the log-transformed total number of COVID-19 cases (defined in **Section 2**, i.e., $\log_{10}(\sum_{j=1}^{31} x_t^j + 1)$, where $x_t^j$ is the monthly cases for a specific province).

Note that population, urbanization rate, and the percentage of age group 1 are not included in the two models because the variances of these variables are tiny after summarizing at the country level. The 12-month lagged fatality and injury metrics are utilized because e-bike safety in China shows a yearly periodicity (see **Fig 3**). Since the two models use the 12-month lagged variable, only 3 years' monthly data with a total of 36 records are utilized for training the models.

## Clustering-based multi-output regression models

For the province-level analysis, we continue with the multi-variate and multi-output settings. The time-series dataset $\boldsymbol{W}$ is collected from $J$ disjoint subpopulations (i.e., provinces and municipalities in this paper) and there are $T$ records (i.e., time periods) for each

subpopulation, i.e., $W = \bigcup\limits_{j=1}^{J} W^j$ and $W^j = \{(\boldsymbol{x}_1^j, \boldsymbol{y}_1^j), (\boldsymbol{x}_2^j, \boldsymbol{y}_2^j), \ldots, (\boldsymbol{x}_T^j, \boldsymbol{y}_T^j)\}$ for $j \in \{1,2,\ldots,J\}$. We group subpopulations into $K$ ($K \leq J$) clusters such that records from one subpopulation shall belong to the same cluster. Let $C_k$, $k \in \{1,2,\ldots,K\}$, denote the subset of data records that belong to cluster $k$. Any two clusters $k$ and $k'$ are disjoint, i.e., $C_k \cap C_{k'} = \emptyset$, for $k \neq k'$ and $k, k' \in \{1,2,\ldots, K\}$; the union of all clusters is the full dataset, i.e., $\bigcup_{k=1}^{K} C_k = W$. In this manner, we cluster provinces based on their similarity in e-bike safety and socioeconomic metrics and the cluster labels of data records from the same province are identical. Furthermore, $\boldsymbol{y}_t^j$ is a two-dimensional vector containing the number of fatalities and number of injuries, $\boldsymbol{x}_t^j$ is an 11-dimensional vector that contains the 12-month lagged number of fatalities, the 12-month lagged number of injuries, population, seasonal GDP, express/logistics profit, and power consumption, average income, urbanization rate, percentage of age group 1, percentage of age group 2, and the log-transformed number of COVID-19 cases. All these variables are directly obtained from the dataset according to specific months and provinces/municipalities.

The clustering task is conducted via the K-means algorithm [36]. Before clustering, one needs to normalize $\boldsymbol{x}_t^j$ and $\boldsymbol{y}_t^j$ based on the general 0–1 approach, i.e., $\tilde{x}_{t,m}^j = \frac{x_{t,m}^j - \min_{j',t'}(x_{t',m}^{j'})}{\max_{j',t'}(x_{t',m}^{j'}) - \min_{j',t'}(x_{t',m}^{j'})}$. Let $\tilde{\boldsymbol{x}}_t^j$, $\tilde{\boldsymbol{y}}_t^j$, and $\tilde{W}^j$ denote the normalized variables and datasets, and let $(\boldsymbol{p}_k, \boldsymbol{q}_k)$ denote the centroid of these normalized data records, where $\boldsymbol{p}_k$ and $\boldsymbol{q}_k$ represent centroids of explanatory variables and dependent variables, respectively. By randomly taking $K$ centroids $\{(\boldsymbol{p}_1, \boldsymbol{q}_1), (\boldsymbol{p}_2, \boldsymbol{q}_2), \ldots, (\boldsymbol{p}_K, \boldsymbol{q}_K)\}$ for initialization, the K-means algorithm iteratively proceeds the following two steps and will terminate once the centroids converge.

- **Assignment Step**. Traverse the subpopulations, i.e., $j \in \{1,2,\ldots,J\}$, for $\tilde{W}^j$, assign all the records to class $k$ if the sum of the squared Euclidean distances to centroid $(\boldsymbol{p}_k, \boldsymbol{q}_k)$ is the minimum for $k \in \{1,2,\ldots,K\}$:

$$L(j, k) = \sum_{t=1}^{T} \left\| (\tilde{\boldsymbol{x}}_t^j, \tilde{\boldsymbol{y}}_t^j) - (\boldsymbol{p}_k, \boldsymbol{q}_k) \right\|^2 \tag{3}$$

$$C_k = \{(\tilde{\boldsymbol{x}}_t^j, \tilde{\boldsymbol{y}}_t^j) \in \tilde{W}^j : L(j,k) \leq L(j,k'), k' \in \{1,2,\ldots,K\}, j \in \{1,2,\ldots,J\}\} \tag{4}$$

where $L(j,k)$ denotes the sum of squared Euclidean distances from each record in subpopulation $j$ to the centroid of cluster $k$.

- **Updating Step.** Traverse the clusters, i.e., $k \in \{1,2,\ldots,K\}$, compute the new centroids based on the current clustering results:

$$\boldsymbol{p}_k = \frac{\sum_{\tilde{W}^j \in C_k} \sum_t^T \tilde{\boldsymbol{x}}_t^j}{\sum_{\tilde{W}^j \in C_k} N^j}, \quad \boldsymbol{q}_k = \frac{\sum_{\tilde{W}^j \in C_k} \sum_t^T \tilde{\boldsymbol{y}}_t^j}{\sum_{\tilde{W}^j \in C_k} N^j} \tag{5}$$

After clustering, the original dataset is divided into $K$ disjointed datasets and data records in the same cluster share a more similar pattern to each other than to those in other clusters. Multi-output regression models are fitted for each specific dataset, which has the following model formulation:

$$\boldsymbol{y}_t^j = \boldsymbol{x}_t^j \sum_{k=1}^{K} I_{j,k} \boldsymbol{w}_k + \boldsymbol{e}_t \tag{6}$$

where $I_{j,k}$ denotes the indicator whether subpopulation $j$ belongs to cluster $k$ (i.e., if $D^j \subseteq C_k$, $I_{j,k}$ = 1; else, $I_{j,k} = 0$), $w_k$ denotes the coefficient matrix for cluster $k$. For each cluster, we estimate $w_k$ by minimizing its regression loss (i.e., Eq (2)).

This novel approach of clustering-based regression will be beneficial in both capturing the heterogeneous effects of explanatory variables on the dependent variables and keep the simplicity of the model. Therefore, we can ascertain the impact of COVID-19 on different clusters of provinces.

## Analysis results

### The overall impact of COVID-19 on e-bike safety in China

We use the multi-output Lasso model (i.e., $\lambda_2 = 0$) for the first research task due to its capability of eliminating collinearity among explanatory variables. For both model non-COVID-19 and model COVID-19, we attempt different values of $\lambda_1$. The results for coefficients and goodness-of-fit are shown in **Table 2**, in which we test the significance of coefficients based on [46] and find that the explanatory variables are significant at 1% to 10% levels. First, we note as $\lambda_1$ increases, the coefficients of more features become zero and the model tends to be simpler due to the elimination of features with a highly collinear relationship; while the goodness-of-fit (i.e., R-Squared, MAE, and RMSE) becomes worse. Second, comparing COVID-19 models with non-COVID-19 models, we observe that the goodness-of-fit notably improves after adding COVID-19 metrics into the model. Consequently, we conclude that COVID-19 significantly affects e-bike safety in China.

Concerning both goodness-of-fit and simplicity (i.e., less collinearity among explanatory variables), we select the COVID-19 model with $\lambda_1 = 1.0$ for quantitative analysis. The lagged number of fatalities and lagged number of injuries are positively related to the outputs, which is consistent with the profile plot in **Fig 3**. The log-transformed number of COVID-19 cases has a strong negative impact on the numbers of fatalities (a coefficient of -11.77) and injuries (a coefficient of -50.85). An intuitive explanation could be that people tend to make fewer trips during the pandemic. Although there are no direct observations on e-bike travel demand, the results are consistent with findings in other modes [29, 33, 35]. Power consumption and seasonal GDP, which reflect living, social, and productivity levels, have positive influences on the

**Table 2. Results of different country-level models.**

| Variables | Non-COVID-19 Model | | | | | | COVID-19 Model | | | | | |
|---|---|---|---|---|---|---|---|---|---|---|---|---|
| | $\lambda_1 = 0.2$ | | $\lambda_1 = 1.0$ | | $\lambda_1 = 5.0$ | | $\lambda_1 = 0.2$ | | $\lambda_1 = 1.0$ | | $\lambda_1 = 5.0$ | |
| | # fata. | # inju. | # fata. | # inju. | # fata. | # inju. | # fata. | # inju. | # fata. | # inju. | # fata. | # inju. |
| intercept | 557.9 | 4925.5 | 462.4 | 4481.0 | 370.2 | 3387.8 | 402.1 | 558.7 | 198.1 | 3562.1 | 209.5 | 2455.0 |
| lagged # of fatalities | 0.217 | 4.893 | 0.238 | 4.558 | 0.199 | 3.456 | 0.205 | 4.661 | 0.269 | 4.832 | 0.240 | 3.943 |
| lagged # of injuries | 0.060 | 0.693 | 0.060 | 0.681 | 0.054 | 0.639 | 0.057 | 0.674 | 0.056 | 0.664 | 0.052 | 0.618 |
| trans. # of COVID-19 | - | - | - | - | - | - | -8.09 | -158.2 | -11.77 | -50.85 | -8.562 | -53.98 |
| seasonal GDP | 0.588 | 3.715 | 1.045 | 3.798 | 0.549 | 4.008 | -0.498 | 7.130 | 0.340 | 0.814 | 0.015 | 0.102 |
| expr./logi. profit | 0.613 | -0.085 | 0.000 | 0.000 | 0.000 | 0.000 | 0.955 | -3.145 | 0.000 | 0.000 | 0.000 | 0.000 |
| power consumption | 0.005 | 1.261 | 0.051 | 1.202 | 0.053 | 1.030 | -0.022 | 1.221 | 0.042 | 1.183 | 0.049 | 1.033 |
| avg. annual income | 5.380 | -134.8 | -2.319 | -123.6 | -4.848 | -86.976 | 1.320 | -304.2 | -4.152 | -144.4 | -4.087 | -95.50 |
| percentage age 2 | -53.79 | -148.9 | -28.07 | -138.9 | -12.35 | -100.1 | -28.76 | 635.4 | 0.000 | 0.000 | 0.000 | 0.000 |
| R-Squared | 0.750 | 0.780 | 0.713 | 0.777 | 0.679 | 0.757 | 0.761 | 0.792 | 0.738 | 0.784 | 0.721 | 0.768 |
| MAE | 20.41 | 228.8 | 21.45 | 229.7 | 22.50 | 241.5 | 20.20 | 212.6 | 21.34 | 225.2 | 21.43 | 232.5 |
| RMSE | 24.91 | 291.1 | 26.71 | 292.1 | 28.25 | 305.8 | 24.39 | 283.1 | 25.50 | 288.9 | 26.35 | 298.8 |

numbers of fatalities and injuries [36]. Due to collinearity between socioeconomic features, we find the coefficients of express/logistics profit and percentage of age group 2 are zero (i.e., the later variable is negatively related to GDP because a higher percentage of aged people means a lower labor level). The average income has small negative coefficients, which could result from the negative relationship between income and e-bike usage (or positive relationship between income and driving/taxi demand) [47]. Given the linear relationship between socioeconomic and e-bike safety metrics, we are unclear whether unsafe riding habits are under control or not during these years [48], so that more variables are needed to conduct further analyses. In summary, COVID-19 relieves e-bike safety issues in China, possibly due to the decline of overall living and social activities and e-bike travel demand.

## Cluster-based analyses of e-bike safety

After trying different numbers of clusters (K) in the subpopulation-based K-means clustering task, we find that two clusters are the best suitable for the fused e-bike safety dataset. Cluster 1 contains 25 provinces/municipalities that are marked with orange dots and numbers in **Fig 2**; while cluster 2 contains the remaining 6 provinces (including Hubei province, one major outbreak place of COVID-19) marked with blue dots and numbers. We show distinctive features in socioeconomic, e-bike safety, and COVID-19 patterns of the two clusters in **Table 3**. The e-bike safety issue concerning the number of fatalities/injuries in cluster 2 is significantly serious than in cluster 1, and the variances of socioeconomic metrics in cluster 2 are smaller than in cluster 1. Furthermore, the usage of e-bikes in provinces in cluster 2 is generally high. For instance, people in Guangxi province have a large demand for e-bike travel because e-bikes are more economical than other modes concerning the relatively low economic level; Jiangsu province is the largest producer and solder for e-bikes in China. Such information can be obtained by googling the keywords.

Following the clustering-based multi-output regression approach introduced in **Section 4**, we try different combinations of $\lambda_1$ and $\lambda_2$ and select $\lambda_1 = 0.1$ and $\lambda_2 = 0$ after balancing goodness-of-fit and model simplicity. The regression results are shown in **Table 4**. First, the impact of (the log-transformed) number of COVID-19 cases on cluster 2 (i.e., -1.660/-21.67 for fatality/injury) is notably higher than cluster 1 (i.e., -0.012/-0.325 for fatalities/injuries). Second, we note the coefficients of population are negative for cluster 2, which is somehow different from previous traffic safety analyses based on small scale (i.e., block-based or zip code-based) populations [37]. This could be caused by the collinearity between population and urbanization

**Table 3. Statistics of two clusters of China provinces.**

| Variables | Cluster 1 | | | Cluster 2 | | |
|---|---|---|---|---|---|---|
| | Min. | Avg. | Max. | Min. | Avg. | Max. |
| # of fatalities for e-bike accidents | 0.000 | 4.927 | 33.00 | 0.000 | 20.95 | 62.00 |
| # of injures for e-bike accidents | 0.000 | 36.07 | 205.0 | 3.000 | 191.6 | 520.0 |
| log-transformed # of COVID-19 cases | 0.000 | 0.312 | 2.919 | 0.000 | 0.298 | 4.776 |
| population (million people) | 0.034 | 0.656 | 3.236 | 0.433 | 1.311 | 2.891 |
| seasonal GDP (trillion CNY) | 0.009 | 1.780 | 22.25 | 0.274 | 2.921 | 13.25 |
| profit from express/logistics (billion CNY) | 0.500 | 17.09 | 116.9 | 10.40 | 28.68 | 70.30 |
| power consumption (billion KWH) | 3.54 | 39.61 | 126.0 | 49.47 | 69.46 | 99.37 |
| urbanization rate | 17.48 | 29.96 | 72.23 | 21.49 | 31.78 | 52.40 |
| average annual income | 30.23 | 61.52 | 86.62 | 50.01 | 58.92 | 68.15 |
| percentage of age group 1 (15 to 64) | 64.47 | 71.30 | 78.81 | 63.37 | 69.10 | 73.32 |
| percentage of age group 2 (over 64) | 5.670 | 11.82 | 17.42 | 10.03 | 13.15 | 16.20 |

Table 4. Regression results of two clusters.

| Variables | Cluster 1 | | Cluster 2 | |
|---|---|---|---|---|
| | # fata. | # inju. | # fata. | # inju. |
| Intercept | 1.492*** | 4.055*** | 26.18*** | 547.1*** |
| lagged # of fatalities | 0.250** | 0.038** | 0.468** | 0.611** |
| lagged # of injuries | 0.025* | 0.886** | 0.002* | 0.301** |
| trans. # of COVID-19 | -0.012** | -0.325** | -1.660** | -21.67*** |
| Population | 0.000 | 0.000 | -0.188** | -0.900** |
| seasonal GDP | 2.071*** | 7.207** | 0.000 | 0.000 |
| expr./logi. profit | 0.000 | 0.000 | -0.306* | -0.045* |
| power consumption | 0.007* | 0.087** | 0.490* | 2.376** |
| urbanization rate | 0.000 | 0.000 | -0.289* | -5.724* |
| avg. annual income | -0.005* | -0.002* | -0.002* | -0.006* |
| % of age group 1 | 0.000 | 0.000 | -0.106** | -1.138* |
| % of age group 2 | 0.000 | 0.000 | 0.000 | 0.000 |
| R-Squared | 0.454 | 0.720 | 0.621 | 0.421 |
| MAE | 2.610 | 12.29 | 6.223 | 42.54 |
| RMSE | 3.728 | 18.42 | 8.217 | 60.69 |

significant levels

* 10%

** 5%

*** 1%

rate, which also has negative coefficients for cluster 2. Meanwhile, the two variables have zero coefficients for cluster 1, indicating that e-bike safety is insensitive to rural and urban populations in provinces with a minor safety issue. Third, e-bike safety is found insensitive to the percentage of age group 2 in both clusters. According to the results in **Table 2**, we know the rate of the aged population is negatively related to e-bike incidents, but the coefficients are dragged to zero with the Lasso regularization. Last, the profit from express/logistics has negative coefficients for cluster 2, this could be caused by the huge online shopping demand in these provinces (Zhejiang and Jiangsu are two giant provinces in China for online shopping) that would reduce the demand for travel as well as for e-bike.

To this end, we have found significant negative impacts of COVID-19 on the numbers of e-bike fatalities and injuries at both the country-level and the clustering-based province-level in China. The fact could be caused by the decline of travel demand due to lockdown policies and the people's panic about the pandemic [19–24]. With the progress of vaccination campaigns, people's travel demand could recover together with e-bike safety issues in the post-pandemic world. Concerning existing and upcoming safety issues, some provinces, such as Guangxi and Jiangsu, have already implemented e-bike "safe-riding policies" (e.g., wearing a helmet is mandatory) in middle and late 2020. Furthermore, the results in this paper could be helpful in policy implications. Our findings suggest that encouraging the express/logistics industry can be a promising way to control e-bike safety accidents. E-bike safety problems could be relieved when people get more income; alternatively, the promotion of other economical travel modes, such as car sharing, could be an effective way for reducing e-bike demand. In addition, for provinces in cluster 2, transportation agencies need to enhance safety enforcement in rural areas.

## Conclusions

This study utilizes novel applications of multi-output linear regression and clustering-based multi-output regression models on e-bike safety analyses. Based on a fused dataset with mainland China's province-level e-bike safety, economic, and COVID-19 metrics, we are particularly interested in the impact of the COVID-19 pandemic on the numbers of e-bike fatalities and injuries in China. We first adopt multi-output regression models to ascertain how COVID-19 affects China's e-bike safety at the country level. The modeling results demonstrate that the pandemic has notably reduced e-bike fatalities and injuries. Second, we use clustering-based multi-output regression models to capture different statistical patterns on the province level. Two clusters of provinces/municipalities are identified: one (cluster 1) with a lower while the other (cluster 2) with higher numbers of e-bike fatalities and injuries. Cluster 2 includes Hubei provinces, which is one major outbreak place of COVID-19. The regression results indicate that the pandemic has a greater negative impact on the number of e-bike fatalities/injuries for provinces in cluster 2, while the influence for provinces/municipalities in cluster 1 is negative but weaker. The regression results can be helpful in policy implications that further reduce e-bike safety issues; for instance, the government could encourage the express/logistics industry to decrease e-bike demand and e-bike accidents.

There are several limitations in this study. First, we acknowledge that there is a lack of monthly e-bike ownership and on-demand delivery information in the existing dataset. As a result, the interplay between COVID-19 and the food (also grocery) delivery industry as well as its impact on e-bike safety are not captured in this paper. Second, the horizon of the current time-series data is short for a comprehensive understanding of how COVID-19 affects e-bike (also other roadway traffic) safety, especially in the post-pandemic world. This is because the pandemic could still coexist with people for a long time and one year's data is not sufficient to fully examine the fundamental changes in mobility and traffic safety patterns. Third, personal travel habit data is not accessible in this paper, which could be important for ascertaining the safe/unsafe driving/riding patterns during the COVID-19 pandemic. Fourth, we haven't involved policies and interventions related to COVID-19 and (or) e-bike regulations; social events such as the "stay-at-home call" could have notable correlations between COVID-19 cases, e-bike travel demand, as well as the overall travel demand across different modes, which also affects the level of safety [2]. The aforementioned limitations will be addressed in future studies once more data becomes available.

## Supporting information

**S1 File.**
(ZIP)

## Acknowledgments

We thank the Research Institute for Road Safety of Ministry of Public Security for data collection. The authors are responsible for all statements.

## Author Contributions

**Data curation:** Xingpei Yan.

**Formal analysis:** Xingpei Yan, Zheng Zhu.

**Funding acquisition:** Xingpei Yan.

**Methodology:** Zheng Zhu.

**Supervision:** Zheng Zhu.

**Visualization:** Xingpei Yan.

**Writing – original draft:** Zheng Zhu.

**Writing – review & editing:** Xingpei Yan.

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
