## [Decision Letter · Decision Letter 0]

27 Jun 2021

PONE-D-21-13700

Quantifying the Impact of COVID-19 on E-Bike Safety in China via Multi-Output and Clustering-Based Regression Models

PLOS ONE

Dear Dr. Zhu,

Thank you for submitting your manuscript to PLOS ONE. After careful consideration, we feel that it has merit but does not fully meet PLOS ONE’s publication criteria as it currently stands. Therefore, we invite you to submit a revised version of the manuscript that addresses the points raised during the review process.

We look forward to receiving your revised manuscript.

Kind regards,

Feng Chen

Academic Editor

PLOS ONE

Journal Requirements:

3. We note that Figure 1 in your submission contain map images which may be copyrighted. All PLOS content is published under the Creative Commons Attribution License (CC BY 4.0), which means that the manuscript, images, and Supporting Information files will be freely available online, and any third party is permitted to access, download, copy, distribute, and use these materials in any way, even commercially, with proper attribution. For these reasons, we cannot publish previously copyrighted maps or satellite images created using proprietary data, such as Google software (Google Maps, Street View, and Earth). For more information, see our copyright guidelines: http://journals.plos.org/plosone/s/licenses-and-copyright.

You may seek permission from the original copyright holder of Figure 1 to publish the content specifically under the CC BY 4.0 license. 

If you are unable to obtain permission from the original copyright holder to publish these figures under the CC BY 4.0 license or if the copyright holder’s requirements are incompatible with the CC BY 4.0 license, please either i) remove the figure or ii) supply a replacement figure that complies with the CC BY 4.0 license. Please check copyright information on all replacement figures and update the figure caption with source information. If applicable, please specify in the figure caption text when a figure is similar but not identical to the original image and is therefore for illustrative purposes only.

Reviewers' comments:

Reviewer's Responses to Questions

**Comments to the Author**

1. Is the manuscript technically sound, and do the data support the conclusions?

Reviewer #1: Yes

Reviewer #2: Partly

2. Has the statistical analysis been performed appropriately and rigorously? 

Reviewer #1: Yes

Reviewer #2: Yes

3. Have the authors made all data underlying the findings in their manuscript fully available?

Reviewer #1: No

Reviewer #2: Yes

4. Is the manuscript presented in an intelligible fashion and written in standard English?

Reviewer #1: Yes

Reviewer #2: Yes

5. Review Comments to the Author

Reviewer #1: The authors provide an interesting study on analyzing the impacts of COVID-19 on e-bike safety in China. By adding/removing COVID-19 covariates, they show that e-bike safety is sensitive to COVID-19. Two clusters of provinces with different sensitivities to COVID-19 are found based on the clustering-based regression method. The topic is timely and the method is sound. Please address the following comments.

1) In Table 1, it seems the max monthly number of fatalities/injuries is only 62/520 for a province. The number seems small since the population and area of a China province are huge. I am not judging the data, but please provide a sufficient explanation on why the numbers are so small? Are they representative?

2) The results need more explanation. In Table 4, the coefficients of "seasonal GDP" and "logit profit" for cluster 2 are negative. The authors claim that "the main reason could be due to the difference in economic structure and people’s commute behavior of the two clusters". Please provide evidence on this. Since readers are not familiar with many provinces in China, more background information in terms of e-bike production/sale/use, economic, population, and geographical is necessary.

3) Following the previous comment, the policies on e-bike usage and COVID-19 control shall also be different across provinces. Will this make an impact on the results of this study? The authors need to discuss this point and obtain more insights. The author also needs to provide some discussions about how the research findings can help improve traffic safety and epidemic control.

4) The authors mention that "30% of road traffic fatalities/injures were caused by e-bike accidents". Is it possible to put a pie chart about the percentage of different traffic accidents in China, this could be interesting and highlight the authors' statement?

5) Please improve the language, now there are some typos and grammar errors.

Reviewer #2: The paper examines the relationship between the number of COVID-19 cases and e-bike safety metrics such as number of fatalities and injuries. This is achieved through fusing multiple data sources and adopting a multi-output clustering-based regression analysis at the national and provincial levels. By interpreting the results of the regressions, the authors conclude that a higher number of covid-19 cases has a negative effect on the number of e-bike fatalities and injuries, and that this effect is more pronounced for the cluster of provinces with more e-bike fatalities and injuries. While the specific subject matter (COVID-19’s impact on e-bike safety) is novel, the paper can be improved by addressing the following issues:

(1) The paper would benefit greatly from a thorough literature review of how COVID-19 affects travel behavior and specifically traffic safety. This will help authors to compare their findings on e-bikes to other modes of transportation like bicycles or motor vehicles. Also, the impact of COVID-19 on logistics and delivery businesses can be examined as well.

(2) Another aspect of a thorough literature review should focus on how socioeconomic factors affect travel volume and safety. This would help justify the choice of predictor variables in the paper, and hopefully support statements like in page 4 line 20-22: “the unbalance in socioeconomic levels brings heterogeneity in the number of COVID-19 cases and the e-bike safety”. Currently, the variable choices are not well justified. Some theoretical foundation is needed here. Other census statistics are also worth trying (even though they are mostly static): population, age, income level, urbanization rate, motorization rate, etc.

(3) K-means typically requires the data (x and y) to be normalized. It is not clear whether this is done. Based on Table 3, it seems the data is not properly normalized. In addition, the K-means is applied to 31 provinces (not 1000+ records), implying that the panel data is somehow aggregated at the provincial level. This needs more explanation.

(4) Some of the “conclusions” in Section 4.1 (page 9 line 13-20) and Section 4.2 (page 10 line 21-32) are mostly speculative and not well supported by model results. The core of the issue lies in how to disentangle the effects of change in overall travel demand vs change in “unsafe riding habits”. Related to (2), a literature review and some theoretical foundation can be helpful here. If this cannot be addressed, then the authors should be very careful about their conclusions.

(5) The writing of the paper needs some refinement. Take Section 2 for example:

• Page 4, line 15-16. The following sentence does not seem to make much sense: “For all these variables, the average values are greater than the medium values, indicating that there are ‘metropolitan’ provinces/municipalities in China.”

• Page 4, line 28: “transferred” -> “log-transformed”? Also, Figure 2 needs a legend.

• Page 5, line 3: “increase” -> “increases”

There are other examples like this. A thorough proofread would be very helpful!

6. PLOS authors have the option to publish the peer review history of their article (what does this mean?). If published, this will include your full peer review and any attached files.

Reviewer #1: No

Reviewer #2: No

---

## [Author Response · Author response to Decision Letter 0]

4 Jul 2021

Please refer to the attached "responses to reviewers" for details

---

## [Decision Letter · Decision Letter 1]

26 Jul 2021

PONE-D-21-13700R1

Quantifying the Impact of COVID-19 on E-Bike Safety in China via Multi-Output and Clustering-Based Regression Models

PLOS ONE

Dear Dr. Zhu,

Thank you for submitting your manuscript to PLOS ONE. After careful consideration, we feel that it has merit but does not fully meet PLOS ONE’s publication criteria as it currently stands. Therefore, we invite you to submit a revised version of the manuscript that addresses the points raised during the review process.

We look forward to receiving your revised manuscript.

Kind regards,

Feng Chen

Academic Editor

PLOS ONE

Journal Requirements:

Reviewers' comments:

Reviewer's Responses to Questions

**Comments to the Author**

1. If the authors have adequately addressed your comments raised in a previous round of review and you feel that this manuscript is now acceptable for publication, you may indicate that here to bypass the “Comments to the Author” section, enter your conflict of interest statement in the “Confidential to Editor” section, and submit your "Accept" recommendation.

Reviewer #1: All comments have been addressed

Reviewer #2: All comments have been addressed

2. Is the manuscript technically sound, and do the data support the conclusions?

Reviewer #1: Yes

Reviewer #2: Yes

3. Has the statistical analysis been performed appropriately and rigorously? 

Reviewer #1: Yes

Reviewer #2: Yes

4. Have the authors made all data underlying the findings in their manuscript fully available?

Reviewer #1: Yes

Reviewer #2: Yes

5. Is the manuscript presented in an intelligible fashion and written in standard English?

Reviewer #1: Yes

Reviewer #2: Yes

6. Review Comments to the Author

Reviewer #1: The authors have addressed all my comments. This quality has been improved substantially. I recommend this manuscript for publication.

Reviewer #2: Overall, the authors did a good job responding to the prior comments. There are only a couple of minor comments:

(1) One of the only remaining issues is the writing. Grammatical errors are still not uncommon. For example, in Literature Review:

- Page 4, Line 22-23, "It was found that as the number of trips dropped greatly and the number of roadway accidents decreased significantly in the US [29-31]"

- Page 5, Line 1-2, "age group and gender is associated with travel mode and driving/riding behavior [39-40]"

- Some wording issues: “lock-down" -> "lockdown"; "over-speed" -> "speeding"

I think the writing can be further improved by another round of proofreading.

(2) It is recommended to replace Fig. 2 with a map showing the provincial boundaries.

(3) In the conclusion section, the authors may consider to add more discussion on policy implications. Currently, it is not clear how we can use the results of this paper in practices.

7. PLOS authors have the option to publish the peer review history of their article (what does this mean?). If published, this will include your full peer review and any attached files.

Reviewer #1: No

Reviewer #2: No

---

## [Author Response · Author response to Decision Letter 1]

28 Jul 2021

Please see the attached response letter for details

---

## [Decision Letter · Decision Letter 2]

11 Aug 2021

Quantifying the Impact of COVID-19 on E-Bike Safety in China via Multi-Output and Clustering-Based Regression Models

PONE-D-21-13700R2

Dear Dr. Zhu,

We’re pleased to inform you that your manuscript has been judged scientifically suitable for publication and will be formally accepted for publication once it meets all outstanding technical requirements.

Kind regards,

Feng Chen

Academic Editor

PLOS ONE

Additional Editor Comments (optional):

Reviewers' comments:

Reviewer's Responses to Questions

**Comments to the Author**

1. If the authors have adequately addressed your comments raised in a previous round of review and you feel that this manuscript is now acceptable for publication, you may indicate that here to bypass the “Comments to the Author” section, enter your conflict of interest statement in the “Confidential to Editor” section, and submit your "Accept" recommendation.

Reviewer #1: All comments have been addressed

Reviewer #2: All comments have been addressed

2. Is the manuscript technically sound, and do the data support the conclusions?

Reviewer #1: Yes

Reviewer #2: Yes

3. Has the statistical analysis been performed appropriately and rigorously? 

Reviewer #1: Yes

Reviewer #2: Yes

4. Have the authors made all data underlying the findings in their manuscript fully available?

Reviewer #1: Yes

Reviewer #2: No

5. Is the manuscript presented in an intelligible fashion and written in standard English?

Reviewer #1: Yes

Reviewer #2: Yes

6. Review Comments to the Author

Reviewer #1: All comments have been successfully addressed by the authors. I recommend the manuscript for publication.

Reviewer #2: The authors have addressed all my prior comments. I have no further comments. I recommend the manuscript for publication.

7. PLOS authors have the option to publish the peer review history of their article (what does this mean?). If published, this will include your full peer review and any attached files.

Reviewer #1: No

Reviewer #2: No

---

## [Editor Report · Acceptance letter]

13 Aug 2021

PONE-D-21-13700R2 

Quantifying the impact of COVID-19 on e-bike safety in China via multi-output and clustering-based regression models 

Dear Dr. Zhu:

I'm pleased to inform you that your manuscript has been deemed suitable for publication in PLOS ONE. Congratulations! Your manuscript is now with our production department. 

Kind regards, 

on behalf of

Dr. Feng Chen 

Academic Editor

PLOS ONE